# Targeted Metabolomics Provide Chemotaxonomic Insights of *Medicago ruthenica*, with Coupled Transcriptomics Elucidating the Mechanism Underlying Floral Coloration

**DOI:** 10.3390/plants11182408

**Published:** 2022-09-15

**Authors:** Lin Zhu, Hongyan Li, Zinian Wu, Zhiyong Li, Maowei Guo, Bu Ning, Wanpeng Liu, Qian Liu, Lei Liu, Zhiyong Wang, Jun Li, Fugui Mi

**Affiliations:** 1College of Grassland and Resources Environment, Inner Mongolia Agriculture University, Hohhot 010010, China; 2Institute of Grassland Research, Chinese Academy of Agricultural Sciences, Hohhot 010010, China; 3Inner Mongolia Autonomous Region Forestry and Grassland Seedling General Station, Hohhot 010010, China; 4Key Laboratory of Herbage & Endemic Crop Biology, Ministry of Education, School of Life Sciences, Inner Mongolia University, Hohhot 010010, China

**Keywords:** *Medicago ruthenica*, alfalfa, chemotaxonomy, flavonoids, anthocyanins, gene expression

## Abstract

*Medicago ruthenica*, a wild legume forage widely distributed in the Eurasian steppe, demonstrates high genetic and phenotypic variation. *M. ruthenica* with a purely yellow flower (YFM), differing from the general phenotype of *M. ruthenica* with a purple flower (PFM), was recently discovered. The similar characteristics of YFM with *Medicago falcata* have led to conflicting opinions on its taxonomy using traditional morphological methods. The lack of chemotaxonomy information about *M. ruthenica* species and the unclear flower coloration mechanisms have hampered their study. Here, we investigated *M. ruthenica* using targeted metabolomics based on the chemotaxonomy method and elaborated the floral coloration mechanisms using transcriptomics. The identified flavonoids were the same types, but there were different contents in YFM and PFM, especially the contents of cyanidin-3-O-glucoside (C3G), an anthocyanin that causes the purple-reddish color of flowers. The over-accumulation of C3G in PFM was 1,770 times more than YFM. Nineteen anthocyanin-related genes were downregulated in YFM compared with their expression in PFM. Thus, YFM could be defined as a variety of *M. ruthenica* rather than a different species. The loss of purple flower coloration in YFM was attributed to the downregulation of these genes, resulting in reduced C3G accumulation. The taxonomic characteristics and molecular and physiological characteristics of this species will contribute to further research on other species with similar external morphologies.

## 1. Introduction

Germplasms of wild relatives are urgently needed for forage crop breeding and increased agricultural food production [1]. *Medicago ruthenica* is a cross-pollinated, diploid (2n = 2x = 16), perennial legume forage crop and a close relative of alfalfa (*Medicago sativa*) [2]. This species is widely distributed in the steppes of Eurasia and has unique physiological mechanisms for responding to environmental stress compared with other *Medicago* species [3]. *M. ruthenica* shows high genetic variation, particularly for stress-resistant genes, thus providing abundant resources for forage crop resistance breeding [4,5]. An alfalfa cultivar (*M. sativa* L. cv. Longmu No. 803), which was obtained by the hybridization of alfalfa (*M. sativa* L. *zhaodong*) and *M. ruthenica*, has been successfully cultivated and exhibits better survival than alfalfa in a cold environment and higher yield [6]. Therefore, the germplasms of wild *M. ruthenica* should be urgently collected for germplasm exploitation and breeding.

The program for the collection of germplasm of *M. ruthenica* species was established in the 1990s in China [4]. Ample natural germplasm resources are available for this species, and phenotypic characteristics vary widely among this germplasm. In the wild, researchers identified the species and obtained germplasm resources using traditional morphology-based methods. Generally, the corolla of *M. ruthenica* is purple-reddish on the outside and yellow-colored on the inside [2,7]. It is interesting to note that the purely yellow corolla types were found by Bu Ning in the second year, after seeds collected in the wild were planted in the field, which is similar to the corolla of *Medicago falcata*. Bu Ning was engaged in forage germplasm collection at the Grassland Research Institute of the Chinese Academy of Agricultural Sciences and collected seeds from the hilly terrain of Shaerqin Town, Hohhot, Inner Mongolia, in the 1990s. Plants always exhibit similar characteristics, and the use of morphological taxonomy gives rise to controversial conclusions [7]. It has also led to discordant opinions that the purely yellow-flowered type should be ascribed to *M. falcata* or it may be a new variant of *M. ruthenica* [8]. The traditional method is not adequate. However, the lack of chemotaxonomy information in *M. ruthenica* species and the unclear flower coloration mechanisms have hampered studies on *M. ruthenica*.

Therefore, in this study, we utilized chemotaxonomy to analyze the chemical difference between purple-flowered and purely yellow-flowered phenotypes of *M. ruthenica* and elaborated on the floral coloration mechanisms using transcriptomics. We aim to provide novel information for the discrimination and classification of *M. ruthenica* species.

## 2. Results

### 2.1. Floral Color Phenotypes

During the investigation of *M. ruthenica*, two distinct petal color phenotypes were observed: one with a purple-reddish color outside and yellow inside the petals (PFM) and the other with purely yellow petals (YFM), as shown in Figure 1. The development and blooming of the flowers take seven days and can be divided into the following stages: initial floret separation and petals packaged in the calyx; the appearance of petals among the calyx lobes; petals emerging from the calyx; more than 2 mm of petals appearing from the calyx but the keel still wrapped by the vexil; and a ready to bloom flower followed by complete blooming at the final stage (Figure 2). During flowering, the floral colors of PFM and YFM were stable.

### 2.2. Sample Quality Control and the Discrimination of Samples in the Validation Set

To identify the potential metabolites related to petal color in *M.*
*ruthenica* species, we used the UPLC-MS-based targeted metabolomics approach for qualitative and quantitative analyses of the flavonoids. PCA and OPLS-DA are widely used in metabolomics for multivariate statistical analysis. PCA is used to analyze all data, highlighting specific samples, whereas OPLS-DA is used for sample grouping and focuses on analyzing differences in grouped samples. Principal component 1 (PC1) and principal component 2 (PC2) explained 96% and 2.3% of the variance, respectively, which separated PFM and YFM (Appendix A). The OPLS-DA value indicated the complete separation of PFM and YFM, suggesting a significant difference in the flavonoid content of PFM and YFM (Appendix A). Therefore, PCA and OPLS-DA are accurate and could be used for further screening of differentially accumulated metabolites (DAMs), particularly flavonoids.

### 2.3. Composition of Anthocyanins and Flavonoids and Concentrations’ Discrepancy in YFM and PFM

The flavonoid metabolites were identified using UPLC-MS, the metabolite data acquisition of samples was performed in both positive and negative ionization modes (Figure 3a), and 48 DAMs were screened based on PCA and OPLS-DA (Figure 3b). There are two metabolic phenotypes in PFM and YFM (Figure 3c). Compared with those in PFM, 25 metabolites were downregulated in YFM, including cyanidin-3-O-glucoside (C3G) and C-glycosylflavones, 14 were upregulated in YFM, and 9 were not significantly different between YFM and PFM (Figure 3c and Appendix A). These DAMs could be considered potential chemical markers for distinguishing the germplasms of PFM and YFM.

As summarized in Table 1, these forty-eight DAMs included twenty flavonoids, six dihydroflavonoids, eleven flavonols, four flavanols, three chalcones, three isoflavones, and one anthocyanin. The types of anthocyanins and flavonoids were identical in YFM and PFM. C3G was the major anthocyanin in YFM and PFM (Table 1). Interestingly, C3G accumulation was reduced in YFM. The over-accumulation of the C3G in PFM was 1770 times more than YFM (Table 1), implying changes in the regulation of anthocyanin synthesis-related genes.

### 2.4. Identification of 3319 Differentially Expressed Genes (DEGs) Using Transcriptome Sequencing

We used transcriptome sequencing to analyze DEGs in purple-reddish and purely yellow-colored *M. ruthenica* flowers to further investigate the molecular basis of the flower color variation. The total RNAs extracted from the petals of both plants were sequenced, yielding 42.11 Gb of clean data (Appendix A). More than 84.36% of the clean reads were mapped to the *M. ruthenica* reference genome [9]. Mean Q20 and Q30 values were ≥98.48% and ≥95.25%, respectively. The GC content was ≥42.48%, indicating high-quality reads that could be used for differential gene expression analysis. PCA of PFM and YFM samples showed that the PC1 contribution was 76.1%, which showed a clear separation of PFM and YFM (Figure 4a).

Gene function was annotated according to the COG, Gene Ontology (GO), Kyoto Encyclopedia of Genes and Genomes (KEGG), KOG, Protein family (Pfam), Swiss-Prot (a manually annotated and reviewed protein sequence database), eggNOG4.5, and National Center for Biotechnology Information non-redundant protein sequences (NR) databases. Overall, 11,614 (21.4%), 36,567 (67.2%), 27,888 (51.3%), 20,285 (37.3%), 31,798 (58.5%), 27,429 (50.4%), 1327 (2.4%), and 45,238 (83.2%) unigenes were matched to the data in the COG, GO, KEGG, KOG, Pfam, Swiss-Prot, eggNOG4.5, and NR databases, respectively (Appendix A).

According to NR annotation, the highest homology matched for PFM and YFM was that for *Medicago truncatula* (85%) (Figure 4b). According to GO classification, 36,566 unigenes were classified into 53 functional terms; 18 terms were categorized in the biological process, 20 in the cellular component, and 15 in the molecular function (Figure 5a). Among them, the groups of the metabolic process (16,979) and cellular process (16,952), cell (17,257) and cell part (17,257), and binding (18,995) and catalytic activity (17,203) were the highest-ranked categories for the biological process, cellular component, and molecular function, respectively (Appendix A).

For the annotation of potential gene functions, we divided these unigenes into 25 categories based on the KOG database, among which 1195 unigenes were annotated into the group of “secondary metabolites biosynthesis, transport, and catabolism” (Figure 5b). Moreover, 20,285 unigenes were matched to 137 KEGG pathways (Appendix A). The most enriched pathway focused on metabolism, particularly the phenylpropanoid biosynthesis pathway. Among them, 491, 158, 4, 70, and 126 key genes were predicted to be involved in the phenylpropanoid, flavonoid, anthocyanin, flavone and flavonols, and isoflavonoid biosynthesis pathways, respectively (Appendix A). In total, 3319 DEGs were identified (Appendix A). Among them, 1261 and 1621 genes were upregulated (Appendix A) and downregulated when comparing YFM with PFM, respectively (Appendix A). According to the KEGG annotations, DEGs involved in flavonoid biosynthesis (19 unigenes) and isoflavonoid biosynthesis (10 unigenes) were downregulated (Figure 6a), and genes involved in flavone and flavonol biosynthesis (9 unigenes) and isoflavonoid biosynthesis (15 unigenes) were upregulated (Figure 6b).

### 2.5. Differential ABP Gene Expression and Their Quantitative Real-Time PCR (qRT-PCR) Validation

We further explored ABP DEGs to elucidate the floral coloration mechanism in PFM and YFM. Anthocyanin synthesis was regulated by the following structural genes—*CHS*, *CHI*, *F3H*, *F3′H*, *DFR*, *ANS*, and *UFGT*. *CHS*, *CHI*, *F3H*, and F*3′H* were upstream genes, and *DFR*, *ANS*, and *UFGT* were downstream genes. A total of 23 DEGs were considered candidate genes (Table 2). Compared with those in PFM, a total of 19 genes (4 *CHS*, 2 *CHI*, 3 *F3H*, 2 *F3′H*, 4 *DFR*, 2 *ANS*, and 2 *UFGT*) were downregulated, and 4 *UFGT* genes were upregulated in YFM. Thus, different ABPs were inferred in PFM and YFM; a schematic model to better understand the formation of pale-yellow *M. ruthenica* flowers is presented in Figure 7.

To verify the transcriptome data, the expression levels of these 23 candidate genes were examined using qRT-PCR (Figure 8); primers used for qRT-PCR are listed in Appendix A. Most of the genes responsible for anthocyanin synthesis were significantly downregulated in YFM, whereas three *UFGT* genes showed significantly higher expression levels in YFM than in PFM, which was consistent with the results of DEG analysis.

## 3. Discussion

Botanical taxonomy is the basis for understanding and classifying plants, exploring plants’ diversity, and utilizing and conserving plant resources. The categorization and phylogeny of plants relying on traditional external morphological characteristics are known to be conflicting [10]. DNA molecular markers have successfully assisted in plant classification [11]. The limitation of this technique is that it can still only partially reveal the relationship among plant species by gene fragments rather than the complete genome [12]. Therefore, other methods need to be employed. 

Notably, the secondary metabolite components are often similar in plants within a taxonomical unit [13]. This indicates that phytochemistry can be used to provide additional evidence for plant taxonomy [14]. With the availability of metabolomics, it has been used generally in *Vicia* and *Siegesbeckiae Herba* species for plant species identification [15,16]. Among the wide variety of secondary metabolites, flavonoids, and anthocyanins are potential chemotaxonomic markers [17]. Flavonoids are natural pigments, and the combinations of flavonoid compounds differ significantly in plant species, thus being responsible for the yellow or white color of flowers [12,18]. Anthocyanins originate from an essential branch of flavonoids’ biosynthesis, and more than 500 anthocyanins can be produced by modifications such as hydroxylation, glycosylation, or methylation [19,20]. The types and content of anthocyanins can also result in flower color polymorphisms that cause variation from pink to red, blue, or purple [21,22]. For example, flowers rich in cyanidin, pelargonidin, or delphinidin display purple, red, or blue petals, respectively [23]. Thus, in this study, we discussed the differences in chemical compounds between purple-flowered and yellow-flowered phenotypes of *M. ruthenica*; we provided information for the classification of *M. ruthenica* species, and elaborated on the floral coloration mechanisms in *M. ruthenica* species.

In total, 48 flavonoid metabolites were identified in *M. ruthenica,* with no difference in the types of metabolites between YFM and PFM (Table 1). These flavonoids could be considered potential chemical markers for discriminating *M. ruthenica* from other species. Among the *Medicago* species, more than 28 flavonoid metabolites in *M. truncatula* were different from those in *M. ruthenica*, including formononetin-7-*O*-glucoside, liquiritigenin, and purpurin [24]. A total of 13 flavonoids in alfalfa were different from those found in *M. ruthenica* such as malvidin 3-*O*-glucoside, petunidin 3-*O*-glucoside, and robinin [25]. Six flavonoid metabolites in *M. falcata* differed from those in *M. ruthenica*, particularly salidroside, laricitrin, and daidzein [26]. These unique-species flavonoid metabolites were not detected in either YFM and PFM. Hongmei et al. [27] found that the chromosome set was highly similar in yellow-flowered *M. ruthenica* and purple-flowered *M. ruthenica* based on karyotype analysis, which suggests a close relationship between them. The study of inter simple sequence repeats also revealed that *M. ruthenica* with purely yellow flowers was closely related to *M. ruthenica* with purple-reddish flowers, rather than *M. falcata* [5]. Hence, *M. ruthenica* with purely yellow flowers could be defined as a variety of *M. ruthenica*.

Among these detected flavonoids, C3G was the major anthocyanin in the *M. ruthenica* species. The over-accumulation of the C3G in PFM was 1,770 times more than YFM (Table 1), which implies changes in the regulation of anthocyanin pathway genes. The relationship between phenotypic variation, metabolic products, and the regulation of anthocyanin pathway genes has been described through studies on colorful flowers [28]. Either a change in gene expression or loss-of-function of the ABP genes could be responsible for the failure of anthocyanidin accumulation, which further impacts the floral color. Compared with PFM, anthocyanin biosynthesis in YFM is blocked by low expression levels of most structural genes, including 4 *CHS*, 2 *CHI*, 3 *F3H*, 2 *F3′H*, 4 *DFR*, 2 *ANS*, and 2 *UFGT* genes, which might disrupt anthocyanin accumulation, leading to the loss of the purple color and the presentation of pale-yellow petals. (Figure 7). Single or multiple ABP genes are downregulated, resulting in reduced anthocyanin accumulation, which has been found in lotus (*Nelumbo nucifera*) with yellow flowers and alfalfa with white flowers [29,30,31]. Thus, the loss of purple flower coloration in YFM was attributed to the downregulation of these genes, resulting in reduced C3G accumulation.

## 4. Materials and Methods

### 4.1. Plant Materials

Germplasms of PFM and YFM were provided by China National Forage Germplasm Bank and planted in the experimental station (with a random plot size) at the Institute of Grassland Research, China Academy of Agricultural Sciences, in 2016. The floral color of these two *M. ruthenica* phenotypes was stable for the entire duration of the study. Petal samples were collected during the flowering and bud phases for targeted metabolomic and transcriptomic analyses, respectively. A total of three biological samples, each from three independent plants, were used in the experiment.

### 4.2. Metabolite Extraction and Detection

To determine total flavonoids, extraction and targeted metabolite profiling were performed by Ollwe gene Technologies Co., Ltd. (Nanjing, China). Briefly, 100 mg powder of each freeze-dried petal sample was extracted using 3 mL of 75% methanol (containing 1% acetic acid). After 30 s of vortexing, the mixture was sonicated at 4 °C for 30 min and centrifuged at 10,000× *g* for 15 min. Finally, the supernatant solution was filtered (0.22 μm pore size) and used for further analysis by ultra-performance liquid chromatography (UPLC). In addition, a quality control sample was prepared by mixing equal aliquots of the supernatants from all of the samples [32].

Analysis of flavonoids was conducted by UPLC-MS using an XCIEX AD system (AB Sciex, Framingham, MA, USA) equipped with an Acuity UPLC BEH C18 column (1.7 μm, 2.1 × 150 mm; Waters Corp., Milford, MA, USA) as previously described by Pang et al. [32]. The solvent gradient was composed of solvent A (1% phosphoric acid), and B (acetonitrile) run at a flow rate of 300 μL/min. A: 0–0.5 min, 90%; 0.5–15 min, 40%; 15–16.01 min, 2%; 16.01–18 min, 2%; 18–18.01 min, 90%; 18.01–20 min, 90%. B: 0–0.5 min, 10%; 0.5–15 min, 60%; 15–16.01 min, 98%; 16.01–18 min, 98%; 18–18.01 min, 10%; 18.01–20 min, 10%. 

The quadrupole time of flight (Q-TOF) profile was assessed in both positive and negative ion modes, and the parameters of ion source were set as the following: curtain gas, 35 psi; ion source gas 1, 55 psi; ion source gas 2, 60 psi; temperature, 500 °C; ion spray voltage, +5000/−4500 V. 

UPLC-MS data analysis was performed as previously described [33]. The relative contents were determined from UPLC-MS peak areas obtained using the MarkerLynx 4.1 software [34]. PCA is used for understanding the relationships among the data matrix, and OPLS-DA is used for calculating the corresponding variable importance in projection values; they were performed to evaluate differentially expressed metabolites using SIMCA 16.0.2 software (Sartorious, Göttingen, Germany). The annotations of the detected metabolites were compared with the laboratory database (Novogene, Beijing, China).

### 4.3. RNA Extraction and Transcriptomic Analysis

Total RNA was extracted from the petal samples using the TRIzol reagent (Invitrogen, Waltham, MA, USA) following the manufacturer’s instructions. The purity, concentration, and integrity of the samples were assessed using Nanodrop, Qubit 2.0 (both from Thermo Fischer Scientific, Waltham, MA, USA) and Agilent 2100 Bioanalyzer (Agilent, Santa Clara, CA, USA), respectively, to guarantee the eligibility of the samples for sequencing. All samples were sent to Biomarker Co., Ltd. (Beijing, China), and six cDNA libraries (including three replicates) were constructed and sequenced using the HiSeq 2000 platform (Illumina, San Diego, CA, USA) and the sequencing by synthesis technique.

Transcriptome assembly was accomplished using the Trinity software, and gene function was annotated based on the NR, Pfam, KOG/COG/eggNOG (Clusters of Orthologous Groups of proteins), Swiss-Prot, KEGG, and GO databases. Gene expression was estimated by RNA-Seq using expectation–maximization, and DEGs were analyzed using DESeq2 (https://bioconductor.org/packages/release/bioc/html/DeSeq2.html, accessed on 30 November 2020).

### 4.4. qRT-PCR

Total RNA was extracted using the method above. Based on the manufacturer’s protocol, the first-strand cDNA synthesis was performed using FastKing gDNA Dispelling RT SuperMix Kit (Tiangen, Beijing, China). The specific primers for qRT-PCR were designed using Primer 5.0 software (PREMIER Biosoft, San Francisco, CA, USA) and synthesized by Sangon Biotech Co., Ltd. (Shanghai, China). qRT-PCR analysis with three biological repeats per sample was performed using the QuantStudio 6 real-time PCR system (Applied Biosystems, Waltham, MA, USA) with 2×SG Fast qPCR Master Mix (Low Rox) (Sangon Biotech, Shanghai, China). *EF-1α* was used as an internal reference gene for normalization. Gene expression was calculated using the 2^−ΔΔCT^ method.

## 5. Conclusions

We found that YFM is closely related to PFM, and *M. ruthenica* with purely yellow flowers could be defined as a variety of *M. ruthenica*. The loss of purple color in YFM with pale-yellow petals can be primarily attributed to ABP gene downregulation, leading to reduced C3G accumulation. The information about flavonoids and floral coloration mechanism provided in this study could be used to classify *M. ruthenica* species. Therefore, studying plants’ taxonomical characteristics combined with their molecular and physiological characteristics will contribute to further research on forage crop species with similar external morphologies and their breeding.

## Figures and Tables

**Figure 1 plants-11-02408-f001:**
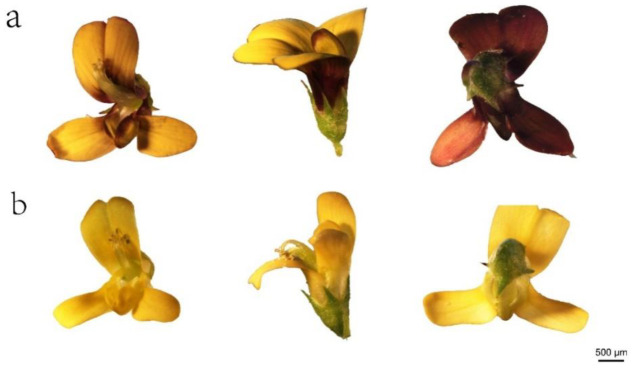
The phenotype observations of the *M. ruthenica* species. The flower samples were collected from PFM (**a**) and YFM (**b**) during the flowering phase at the same time of day (11–12 am) and stored at 4 °C to keep them fresh; then, the samples were inspected under a stereomicroscope and photographed. Flower face (left), side (middle), and bottom (right) images are shown. During the investigation of *M. ruthenica*, two distinct flower color phenotypes were observed, with one having the purple-reddish outside and yellow petal inside (PFM) and the other with the purely yellow petal (YFM).

**Figure 2 plants-11-02408-f002:**
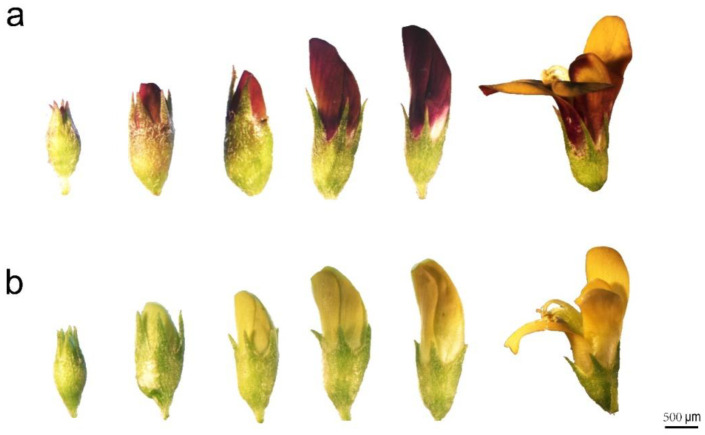
Development observations of *M. ruthenica* with different floral color phenotypes. The samples of different flower development stages were collected from PFM (**a**) and YFM (**b**) and stored at 4 °C to keep them fresh. The samples were then inspected under a stereomicroscope and photographed. It takes seven days for the development and blooming of the flower tissue. The developmental process from the stage of floret includes initial separating and calyx packaging the petals, the stage of petals appearing among the calyx lobes, the stage of the petals exceeding the calyx, the stage of the petals exceeding the calyx more than 2 mm but with the vexil still wrapped by the keel, the stage of the flower ready to bloom, and finally the flower in full bloom, shown from left to right. The floral colors of PFM and YFM were stable.

**Figure 3 plants-11-02408-f003:**
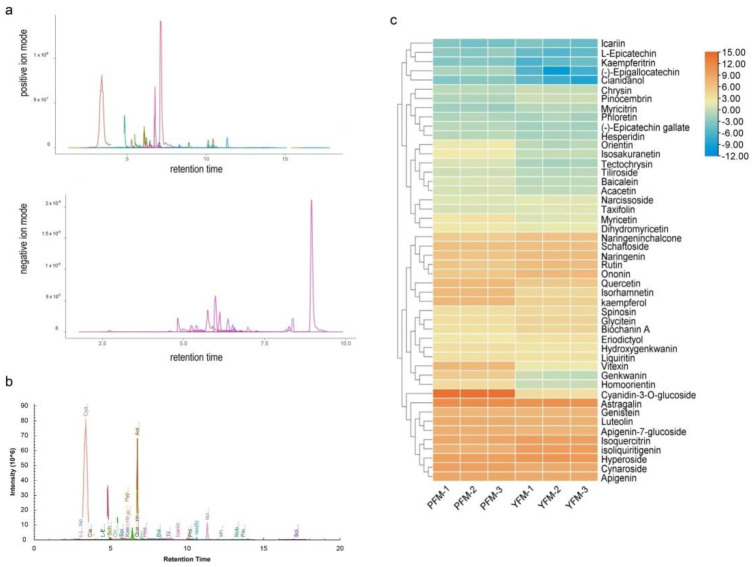
Identification of flavonoids in *M. ruthenica* species. The petal samples of PFM and YFM were collected during the flowering phase, and a total of three biological samples were taken, each from three independent plants that were used in the experiment. To determine the total flavonoids, extraction and targeted metabolite profiling were performed by Ollwe gene Technologies Co., Ltd. (Nanjing, China). (**a**,**b**) Ultra-performance liquid chromatography–tandem mass spectrometry (UPLC-MS) analysis of anthocyanins and flavonoids in flower samples of PYM and YFM. UPLC was performed by XCIEX AD system (AB Sciex, Framingham, MA, USA), and MS performed by QTrap 6500 (AB Sciex, Framingham, MA, USA), UPLC was used for component separation in samples, MS was used for signal acquisition and these separated components’ identification. The signals of these DAMs were acquired and identified in both positive ion mode and negative ion mode (**a**). Forty-eight DAMs were screened based on PCA and OPLS-DA (**b**). In this figure, each peak represents a kind of anthocyanin or flavonoid. The contents of anthocyanins and flavonoids were calculated using the area of related peak. (**c**) Heatmap of differentially accumulated flavonoid metabolites (DAMs) in PFM and YFM. A heatmap was drawn with TBtools using log^2^ fold-change values of the concentrations of these 48 DAMs. Each line presents the color group representing related metabolites, which are listed on the right. The colors reflect the contents of metabolites, cells with orange-color represent high accumulation in samples, and the color scale from blue to red represents low to high accumulation in samples, respectively. PFM−1–3 and YFM−1–3 represent the three biological independent PFM and YFM samples, respectively. In this study, PFMs served as the control group.

**Figure 4 plants-11-02408-f004:**
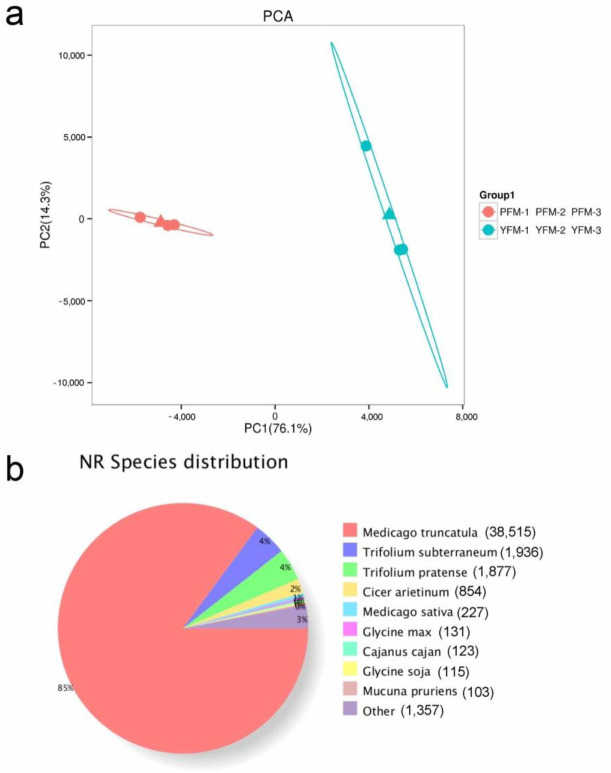
The preliminary statistics analysis (PCA) of RNA-sequencing. (**a**) PCA was performed on all RNA-seq samples. Petal samples in the bud phase were collected for transcriptomic analyses. PFM−1–3 and YFM−1–3 represent the three biological independent PFM and YFM samples, respectively. Total RNA was extracted from the petal samples using the TRIzol reagent (Invitrogen, Waltham, MA, USA) following the manufacturer’s instructions. All samples were sent to Biomarker Co., Ltd. (Beijing, China), and six cDNA libraries (including three replicates) were constructed and sequenced using the HiSeq 2000 platform (Illumina, San Diego, CA, USA) and the sequencing by synthesis (SBS) technique. The contributions of principal component 1 (PC1) and principal component 2 (PC2) were 76.1% and 14.3%, respectively. The result of PCA revealed that the six samples could be assigned to two groups. (**b**) NR annotation of unigenes of PFM and YFM. In comparison with other species, these unigenes of PFM and YFM showed the highest similarity with sequences from *Medicago truncatula* (85%, 38,515), *Trifolium subterraneum* (4%, 1936), and *Trifolium pratense* (4%, 1877).

**Figure 5 plants-11-02408-f005:**
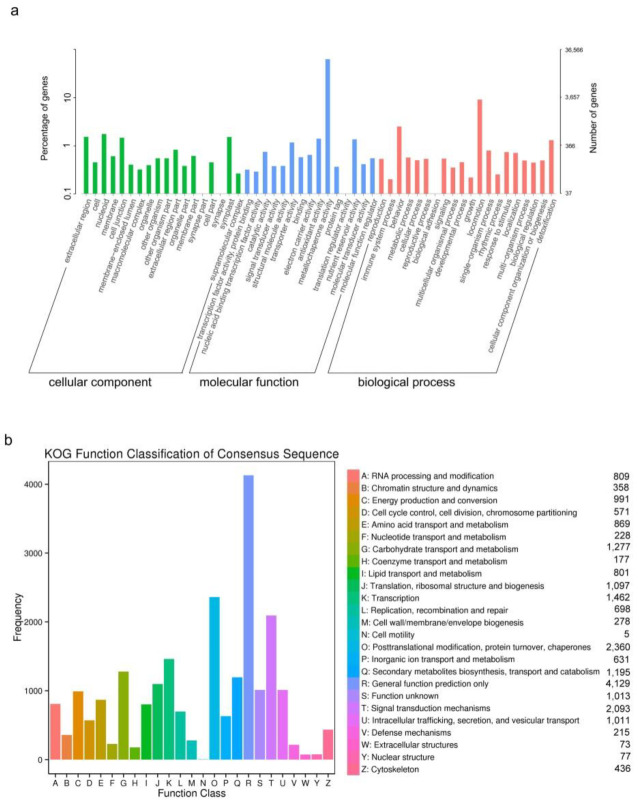
The annotation of PFM and YFM unigenes based on GO (**a**) and KOG databases (**b**). (**a**) GO annotation. The GO annotation was used to classify the possible functions of the unigenes based on NR annotation. The *X*-axis represents the category of the GO function. The *Y*-axis on the left indicates the percentage of the total number of unigenes. The *Y*-axis on the right shows the number of unigenes common to the corresponding GO function. In total, 36,566 unigenes were classified into 53 functional terms; the biological process, cellular component, and the molecular function categories include 19, 20, and 15 GO terms, respectively. (**b**) KOG annotation of unigenes. The *X*-axis is the categories of the KOG annotation, capital letters represent related categories and are listed on the right. The *Y*-axis on the left indicates the total number of unigenes and the specific unigene numbers are marked after related categories. For KOG annotation, these unigenes were annotated into 25 categories, and 1195 unigenes were annotated into the group of “secondary metabolites biosynthesis, transport, and catabolism”.

**Figure 6 plants-11-02408-f006:**
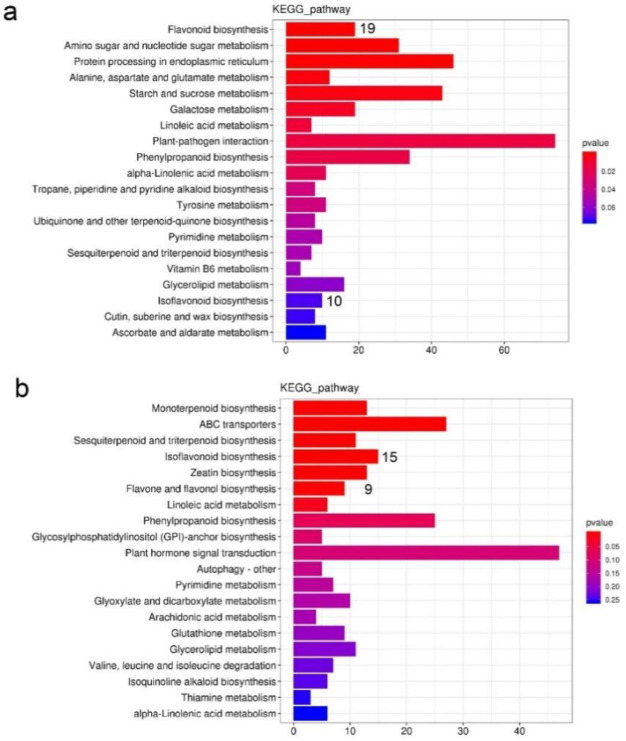
KEGG enrichment analysis of the differentially expressed genes (DEGs). In this study, PFMs served as the control group. (**a**) KEGG analysis of downregulated genes from the RNA-seq data. The *Y*-axis on the left represents the categories of the KEGG annotation, and the *X*-axis indicates the number of downregulated genes common to the corresponding KEGG annotation. In total, the number of downregulated genes involved in categories of flavonoid biosynthesis and isoflavonoid biosynthesis was 19 and 10, respectively. (**b**) KEGG analysis of upregulated genes from the RNA-seq data. The *Y*-axis on the left represents the categories of the KEGG annotation, and the *X*-axis indicates the number of upregulated genes that are common to the corresponding KEGG annotation. The number of upregulated genes involved in categories of flavone and flavonol biosynthesis and isoflavonoid biosynthesis was 9 and 15, respectively.

**Figure 7 plants-11-02408-f007:**
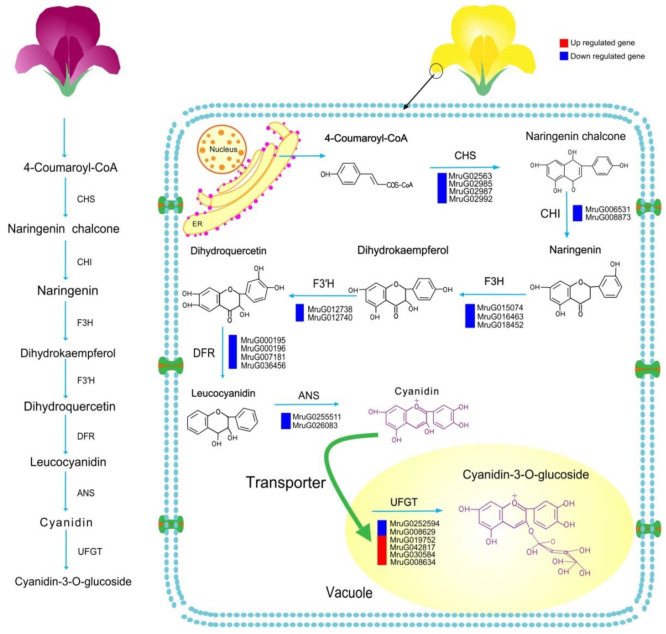
Model representing the anthocyanin synthesis process in the *M. ruthenica* species. In this study, PFMs served as the control group. The referred model for a better understanding of the formation of the pale-yellow *M. ruthenica* flower was drawn using Adobe Illustrator 2019. ER represents the endoplasmic reticulum. The crucial candidate gene IDs are indicated at the side of related enzymes. Blue-colored grids indicate the downregulated genes, and red-colored grids indicate upregulated genes. The arrow refers to the steps of the anthocyanin biosynthesis pathway (ABP). *CHS*, *CHI*, *F3H*, and *F3′H* were upstream genes involved in the ABP. *DFR*, *ANS*, and *UFGT* were downstream genes involved in the ABP. Briefly, the low expression level of 4 CHS, 2 CHI, 3 *F3H*, 2 *F3′H*, 4 *DFR*, 2 *ANS*, and 2 *UFGT* might disrupt the anthocyanin synthesis, leading to the formation of the pale-yellow petals seen in YFM. *UFGT* catalyzes the last step of the ABP, the *UFGT* enzyme glycolyzed anthocyanidin into anthocyanin by adding sugar moieties, thereby increasing the hydrophilicity and stability of anthocyanin.

**Figure 8 plants-11-02408-f008:**
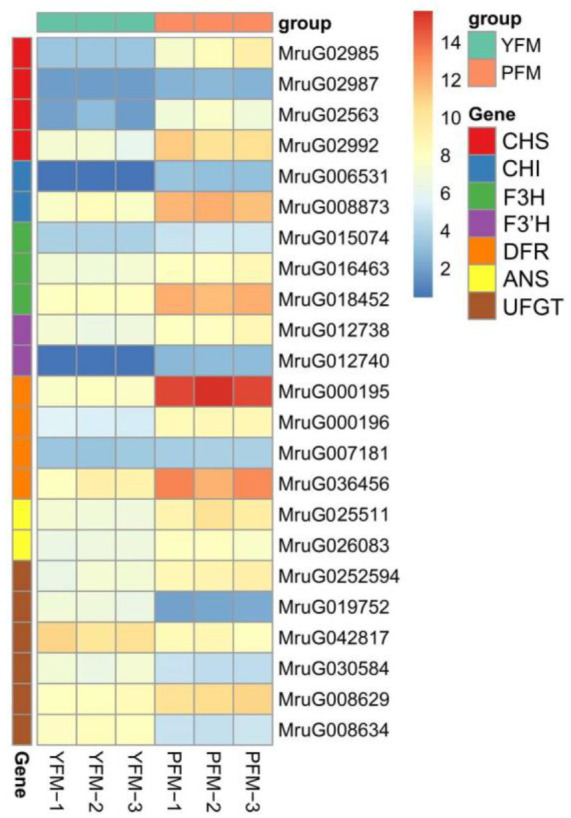
Expression profiles of 23 candidate genes and qRT-PCR validation with the heatmap. Heatmaps of differentially expressed genes’ (DEGs) validation in PFM−1 vs. YFM-1, PFM−2 vs. YFM−2, and PFM−3 vs. YFM−3. In this study, PFMs served as the control group, and *EF-1α* was used as the internal reference gene for normalization. Gene expression was calculated using the 2^−ΔΔCT^ method. The color grids on the left with red, blue, green, orange, yellow, and brown represent *CHS*, *CHI*, *F3H*, *F3′H*, *DFR*, *ANS*, and *UFGT*, respectively. The top grids represent samples of PFM (green) and YFM (orange), PFM−1–3 and YFM−1–3 represent the three biological independent PFM and YFM samples, respectively. The gene names and related gene IDs are listed on the right. The different colors, from blue (low) to red (high), show the expression of the 23 DEGs.

**Table 1 plants-11-02408-t001:** Differentially accumulated flavonoid metabolites in YFM and PFM.

NO.	Metabolite	Types	Polarity	Rt (min)	Related Contents (μmol/g)						*p*-Value	FC (YFM/PFM)
					YFM-1	YFM-2	YFM-3	PFM-1	PFM-2	PFM-3		
1	Cyanidin-3-O-glucoside	anthocyanidins	1	3.36	12.96	11.15	9.77	18,153.86	19,748.49	18,996.23	0.001	0.0006
2	(-)-Epigallocatechin	flavanols	1	3.07	0.02	0.00	0.01	0.35	0.39	0.39	0.000	0.0225
3	Genkwanin	flavonoids	1	13.64	1.22	0.93	0.85	38.05	41.18	40.75	0.025	0.0250
4	Vitexin	flavonoids	1	5.91	4.03	4.05	4.25	134.06	135.03	139.94	0.030	0.0302
5	Homoorientin	flavonoids	1	5.03	1.06	1.17	1.20	13.13	12.95	14.06	0.085	0.0854
6	Cianidanol	flavanols	1	3.64	0.03	0.01	0.00	0.11	0.13	0.10	0.105	0.1046
7	Isorhamnetin	flavonols	1	10.64	18.05	18.21	19.49	122.01	122.26	120.99	0.153	0.1526
8	Orientin	flavonoids	1	5.32	0.71	0.49	0.72	3.61	4.51	3.67	0.163	0.1633
9	L-Epicatechin	flavanols	1	4.5	0.04	0.02	0.03	0.11	0.19	0.14	0.193	0.1934
10	Isosakuranetin	dihydroflavonoids	1	13.52	0.93	0.95	0.94	4.77	4.87	4.93	0.194	0.1938
11	kaempferol	flavonols	1	10.37	28.23	28.48	28.37	141.99	144.90	141.06	0.199	0.1988
12	Tectochrysin	flavonoids	1	16.86	0.47	0.35	0.38	1.85	2.01	1.92	0.207	0.2068
13	Kaempferitrin	flavonols	1	6.1	0.01	0.03	0.04	0.11	0.09	0.09	0.281	0.2807
14	Myricetin	flavonoids	1	7.4	2.07	2.54	2.85	8.69	9.46	6.24	0.305	0.3054
15	Quercetin	flavonols	−1	8.95	44.20	44.90	46.54	134.23	137.88	128.82	0.338	0.3383
16	Baicalein	flavonoids	−1	8.15	0.68	0.82	0.78	2.06	1.86	1.96	0.388	0.3885
17	Cynaroside	flavonoids	1	6.21	339.96	354.97	358.97	748.84	717.89	754.48	0.474	0.4745
18	Tiliroside	flavonoids	1	8.83	0.67	0.57	0.69	1.41	1.340	1.26	0.481	0.4809
19	(-)-Epicatechin gallate	flavanols	−1	6.1	0.30	0.35	0.29	0.59	0.55	0.64	0.528	0.5280
20	Acacetin	flavonoids	1	13.39	0.84	0.86	0.90	1.51	1.70	1.65	0.534	0.5340
21	Hesperidin	dihydroflavonoids	−1	7.24	0.32	0.41	0.38	0.65	0.70	0.60	0.567	0.5669
22	Dihydromyricetin	flavonols	1	4.81	3.08	3.34	3.43	5.75	5.62	5.61	0.579	0.5792
23	Hydroxygenkwanin	flavonoids	1	12.08	6.75	6.73	6.76	11.52	11.23	11.32	0.594	0.5939
24	Luteolin	flavonoids	1	8.9	205.21	206.01	214.63	319.76	319.67	326.94	0.648	0.6476
25	Apigenin-7-glucoside	flavonoids	1	7.05	179.48	183.14	186.88	260.15	248.46	261.00	0.708	0.7076
26	Astragalin	flavonols	1	6.75	1835.43	1902.50	1880.48	2573.41	2492.87	2579.29	0.735	0.7349
27	Icariin	flavonoids	1	9.42	0.09	0.05	0.04	0.06	0.06	0.11	0.753	0.7526
28	Phloretin	chalcone	1	10.27	0.35	0.52	0.30	0.56	0.47	0.45	0.789	0.7888
29	Apigenin	flavonoids	1	10.13	363.68	357.08	379.27	448.01	456.86	482.19	0.794	0.7931
30	Genistein	flavonoids	1	10.13	141.81	143.86	150.57	175.34	178.58	187.64	0.806	0.8055
31	Schaftoside	flavonoids	1	4.97	78.80	77.17	83.68	77.88	81.38	83.61	0.987	0.9867
31	Liquiritin	dihydroflavonoids	1	6.13	9.60	9.86	9.72	10.07	9.94	9.23	1.00	0.9975
33	Narcissoside	flavonoids	−1	6.56	2.94	2.74	2.71	2.32	2.27	2.33	1.21	1.2121
34	Taxifolin	flavonols	1	6.35	2.17	2.32	2.28	1.73	1.94	1.82	1.24	1.2351
35	Chrysin	flavonoids	1	13.34	1.14	0.98	0.98	0.67	0.62	0.66	1.59	1.5868
36	Spinosin	flavonoids	−1	5.75	17.89	16.45	17.06	10.47	10.40	11.16	1.61	1.6050
37	Eriodictyol	dihydroflavonoids	1	8.85	8.68	8.95	9.23	5.49	5.39	5.48	1.64	1.6421
38	Isoquercitrin	flavonols	1	6.08	784.94	706.57	695.53	341.99	490.93	419.78	1.75	1.7459
39	Naringenin	dihydroflavonoids	1	10.18	100.29	99.78	102.12	55.23	56.57	58.11	1.78	1.7786
40	Naringenin chalcone	chalcone	1	10.17	64.49	65.83	65.93	34.79	35.45	37.02	1.83	1.8299
41	Hyperoside	flavonols	1	6.07	1286.82	1385.39	1189.61	652.77	740.89	685.34	1.86	1.8575
42	Glycitein	isoflavones	1	8.77	20.60	21.39	21.02	10.92	11.21	11.09	1.90	1.8968
43	Biochanin A	isoflavones	1	13.93	20.79	20.01	20.95	9.29	9.58	9.71	2.16	2.1608
44	Rutin	flavonols	1	5.79	112.67	122.59	107.68	43.37	44.62	55.40	2.39	2.3916
45	Myricitrin	flavonols	−1	6.07	0.58	0.63	0.64	0.25	0.23	0.25	2.53	2.5284
46	Ononin	flavonoids	1	8.3	123.76	124.57	124.68	51.60	47.81	45.40	2.58	2.5759
47	Pinocembrin	dihydroflavonoids	1	13.68	1.34	1.19	1.31	0.46	0.45	0.46	2.81	2.8057
48	Isoliquiritigenin	chalcone	1	11.34	789.70	788.05	797.03	196.61	197.93	203.57	3.97	3.9705

FC represents the fold-change and the multiple relationship of the substance after the comparison between the two groups. Rt represents retention time. Related contents were calculated using the area of mass spectrum peaks. Forty-eight differentially accumulated flavonoid metabolites (DAMs) were identified, comprising twenty flavonoids, six dihydroflavonoids, eleven flavonols, four flavanols, three chalcones, three isoflavones, and one anthocyanin. The signals of these DAMs were acquired and identified in both positive ion mode and negative ion mode. In column polarity, “1” represents positive ion mode and “−1” represents negative ion mode. The contents of 25 metabolites were lower in YFM than in PFM, especially that of cyanidin-3-O-glucoside.

**Table 2 plants-11-02408-t002:** Selection of key differentially expressed genes of anthocyanin synthesis pathway in *M. ruthenica*.

Gene	Gene ID	FPKM			YFM-FPKM (Mean)	FPKM			PFM-FPKM (Mean)	log^2^FC	Regulation
		YFM-1	YFM-2	YFM-3		PFM-1	PFM-2	PFM-3			
*CHS*	MruG02985	45.409	41.605	42.382	43.132	96.761	95.369	94.306	95.479	−1.0933	down
*CHS*	MruG02563	8.449	8.315	9.041	8.602	22.411	20.558	21.163	21.377	−1.2513	down
*CHS*	MruG02992	0.907	0.749	0.582	0.746	6.559	4.906	2.970	4.812	−2.3438	down
*CHS*	MruG02987	11.239	7.972	8.767	9.326	262.238	272.324	267.033	267.198	−4.7440	down
*CHI*	MruG008873	1.740	1.866	0.923	1.510	3.450	3.373	3.136	3.320	−1.0093	down
*CHI*	MruG006531	134.095	146.950	156.917	145.988	311.336	298.964	306.483	305.595	−1.0119	down
*F3H*	MruG015074	0.000	0.000	0.000	0.000	2.918	2.166	1.886	2.323	−4.3940	down
*F3H*	MruG016463	2.274	2.341	2.052	2.222	13.713	13.612	14.737	14.021	−2.5314	down
*F3H*	MruG018452	5.761	4.335	7.982	6.026	23.731	25.329	25.807	24.956	−1.9320	down
*F3′H*	MruG012738	3.721	3.505	3.884	3.704	11.608	14.313	8.023	11.315	−1.4392	down
*F3′H*	MruG012740	4.021	2.984	2.926	3.310	11.552	9.667	9.268	10.162	−1.4914	down
*DFR*	MruG000195	0.000	0.000	0.000	0.000	188.822	185.284	185.727	186.611	−10.2740	down
*DFR*	MruG000196	0.318	0.524	0.489	0.444	145.809	143.038	142.730	143.859	−7.9463	down
*DFR*	MruG007181	25.204	25.597	27.086	25.962	52.308	57.399	54.999	54.902	−1.0256	down
*DFR*	MruG036456	2.354	1.770	2.822	2.316	5.361	5.806	5.354	5.507	−1.1270	down
*ANS*	MruG0255511	0.820	0.930	0.960	0.903	1.880	1.850	1.910	1.880	−1.9634	down
*ANS*	MruG026083	0.685	0.169	0.242	0.365	434.075	442.808	436.607	437.830	−7.8978	down
*UFGT*	MruG0252594	1.880	2.400	2.320	2.200	3.620	2.910	2.990	3.173	−2.0093	down
*UFGT*	MruG008629	1.442	1.562	1.856	1.620	4.622	4.555	4.506	4.561	−1.3855	down
*UFGT*	MruG042817	11.856	10.709	10.530	11.032	0.225	0.190	0.240	0.218	5.1867	up
*UFGT*	MruG030584	1.283	1.288	1.260	1.277	0.832	0.446	0.440	0.573	1.0296	up
*UFGT*	MruG008634	6.416	5.818	6.750	6.328	1.782	1.839	1.705	1.775	1.8024	up
*UFGT*	MruG019752	2.039	2.417	3.234	2.563	1.293	0.313	0.794	0.800	1.4574	up

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
