# Peer review of "Targeted Metabolomics Provide Chemotaxonomic Insights of Medicago ruthenica, with Coupled Transcriptomics Elucidating the Mechanism Underlying Floral Coloration"

_plants, 2022, doi:10.3390/plants11182408_

Round 1
Reviewer 1 Report
Medicago ruthenica is an important forage crop and, recently, has been recognized as a source of genes to improve abiotic stress tolerance in the Medicago species. Here, the authors investigated on a particular variety of M. ruthenicawhich differs from the canonical one for the color of the flowers, yellow instead of purple-red. They used a metabolomic approach coupled to transcriptomic analysis in order identify the biochemical/molecular basis for defining that both belong to the same species.
The work is interesting and the experimental design is well done, but it presents some weak points, which have to be addressed before the work can be accepted.
Major points
Line 110: I cannot find any details about the identification of the metabolites described. Please, add a table with the detected masses and the identification level used.
Table 1: I suggest to add a column with the p-value for better understanding the results discussed.
Minor points:
Line 25 and line 120: I suggest to change “… 0.0006 times significantly in YFM higher..” with something that highlights the overaccumulation of the C3G in PFM more than 1770 times.
Figure 3, line 142: UPLC and..? What is the detector used for the chromatogram shown in the figure?
Line 158 and line 215: use italic for Figure 4 and 6
Author Response
1 Line 110: I cannot find any details about the identification of the metabolites described. Please, add a table with the detected masses and the identification level used.
Here I added a picture named Fig 3a and changed original Fig 3a,3b into Fig 3b and 3c. The newly added Fig 3a was represented the extracted ion chromatograms of metabolites in both positive and negative ionization modes. I also changed Fig 3b (original Fig 3a), forty-eight DAMs were screened based on PCA and OPLS-DA (b), here I added the related names of differentially accumulated metabolites to each peak.
2 Table 1: I suggest to add a column with the p-value for better understanding the results discussed.
I added a column with the p-value in Table 1
3 Line 25 and line 120: I suggest to change “… 0.0006 times significantly in YFM higher..” with something that highlights the overaccumulation of the C3G in PFM more than 1770 times.
I have already changed the sentences in Line 25, line121 and line 120
4 Figure 3, line 142: UPLC and..? What is the detector used for the chromatogram shown in the figure?
Ultra-performance liquid chromatography Tandem Mass Spectrometry (UPLC-MS) method was used for analysis the anthocyanins and flavonoids.
Ultra-performance liquid chromatography (UPLC) was performed by XCIEX AD (AB Sciex, Framingham, MA, USA) that equipped with an Acuity UPLC BEH C18 column (1.7 μm, 2.1 × 150 mm; Waters Corp., Milford, MA, USA), and Mass Spectrometry (MS) was used QTrap 6500 (AB Sciex, Framingham, MA, USA).
UPLC was used for separation of the components in samples, MS was used for signal acquisition and identifying these components
5 Line 158 and line 215: use italic for Figure 4 and 6
I amended ‘Medicago ruthenic' italic

Reviewer 2 Report
The paper can be published after minor revision.
1. In table 1 47 flavonoids were identified. Table 1 lack of number for each metabolite. It is not reported if any metabolites was acquired in UPLC-MS in negative ion modo or in positive ion mode or both.
2. Figure 3a. I think the UPLC-MS profile need to be in a bigger figure showing all the metabolites detected. In addition authors shoud present both results in positive and in negative ion mode.
3. All over the text and figure captions. Medicago ruthenica should be in italic ( it is not for example in line 69, 79, 87... etc. etc.)
4. Table 1. Names of compounds : there are same names with small initial character. It is necessari to standardize typing.
Author Response
1. In table 1 47 flavonoids were identified. Table 1 lack of number for each metabolite. It is not reported if any metabolites was acquired in UPLC-MS in negative ion mode or in positive ion mode or both
Here I added two columns named NO. and Polarity in the table 1,trying to amend the problems.
2. Figure 3a. I think the UPLC-MS profile need to be in a bigger figure showing all the metabolites detected. In addition authors shoud present both results in positive and in negative ion mode.
Here I added a picture named Fig 3a and changed original Fig 3a,3b into Fig 3b and 3c. The newly added Fig 3a was represented the extracted ion chromatograms of metabolites in both positive and negative ionization modes. I also changed Fig 3b (original Fig 3a), forty-eight DAMs were screened based on PCA and OPLS-DA (b), I added names of flavonoids related to each peak in FIG 3c.
3. All over the text and figure captions. Medicago ruthenica should be in italic ( it is not for example in line 69, 79, 87... etc. etc.)
I feel so sorry, here I amended Medicago ruthenica in italic
4. Table 1. Names of compounds : there are same names with small initial character. It is necessari to standardize typing.
I amended isoliquiritigenin into Isoliquiritigenin
